# Identification of Chaotic Dynamics in Jerky-Based Systems by Recurrent Wavelet First-Order Neural Networks with a Morlet Wavelet Activation Function

Daniel Alejandro Magallón-García [1,2] , Luis Javier Ontanon-Garcia [2,*] , Juan Hugo García-López [3] , Guillermo Huerta-Cuéllar [3] and Carlos Soubervielle-Montalvo [4]

[1] Preparatoria Regional de Lagos de Moreno, Universidad de Guadalajara, Lagos de Moreno 47476, Mexico
[2] Coordinación Académica Región Altiplano Oeste, Universidad Autónoma de San Luis Potosí, Salinas 78600, Mexico
[3] Optics, Complex Systems and Innovation Laboratory, Centro Universitario de los Lagos, Universidad de Guadalajara, Lagos de Moreno 47463, Mexico
[4] Faculty of Engineering, Universidad Autónoma de San Luis Potosí, San Luis Potosí 78280, Mexico
[*] Correspondence: luis.ontanon@uaslp.mx; Tel.: +52-(496)-963-4030

**Abstract:** Considering that chaotic systems are immersed in multiple areas of science and nature and that their dynamics are governed by a great sensitivity to the initial conditions and variations in their parameters, it is of great interest for the scientific community to have tools to characterize and reproduce these trajectories. Two dynamic chaotic systems whose equations are based on the jerky system are used as benchmarks, i.e., the Memristive Shaking Chaotic System (MSCS) and the Unstable Dissipative System of type I (UDSI). One characteristic common to them is their simple mathematical structure and the complexity of their solutions. Therefore, this paper presents a strategy for identifying chaotic trajectories using a recurrent wavelet first-order neural network (RWFONN) that is trained online with an error filtering algorithm and considering the Morlet-wavelet as an activation function. The parameters of the network are adjusted considering the Euclidean distance between the solutions. Finally, the results depict proper identification of the chaotic systems studied through analysis and numerical simulation to validate the behavior and functionality of the proposed network.

**Keywords:** dynamic systems; chaos theory; artificial neural network; error filter algorithm; Morlet-wavelet activation function

**MSC:** 34H10; 70K99; 93B30; 93C10





## 1. Introduction

Chaotic systems are of great importance to the scientific community today given the complexity that can occur in their trajectories over time. Furthermore, in nature, they are present in multiple areas [1]. Current examples of applications of these systems, to mention a few, are: in the generation of chaotic trajectories mobile robots [2] and secure information transmission in cryptography [3,4], in the search algorithms in computer technology utilizing chaotic lattices [5], and in the evolution of convolutional neural network architectures for the classification of brain tumors using magnetic resonance imaging [6].

For those reasons, in recent decades, particular interest has been placed on the design of strategies for the generation of complex trajectories [7], analysis of the characteristics of chaotic solutions [8,9], multistability prediction in chaotic systems [10], and the design of control methods for the generation and suppression of chaotic behavior [2], among other topics related to the study of their behavior—for example, in the synchronization and coupling of states [11,12] and the generation and analysis of time series to implement them for modulation schemes or encrypt them in communication systems [13].

Some interesting systems to study in this sense are those whose equations are defined by the jerky equation, and their dynamics are governed by a commutation or control law dependent on the position of one of its states, for example, in the case of the unstable dissipative systems of type I [14], which are linear systems based on the jerky equation whose only non-linearity is given by a linear affine vector that changes value depending on the position of the state $x$. This type of system has the characteristic that depending on the commutation law that is used, it is possible to obtain solutions that present a complex multi-scrolling behavior in the projection of their phase states. In a similar sense, the simple characterization of the second-order system based also on the jerky equation of the second degree, with the introduction of a control non-linearity based on the memristor behavior, is proposed in [15]. This system, which was found with an exhaustive numerical searching algorithm, presents a complex behavior with three degrees of freedom that results in chaotic behavior and a strange attractor. A common feature in both systems is that they can be implemented electronically for the physical verification of their real complex states, as presented in [8,15].

Regarding the identification of characteristics in dynamic systems, much progress has been made today thanks to the new proposals and methods of artificial intelligence. A characteristic that is important to consider in this sense is the evolution in time of these systems, since although their equations may not present an explicit dependence on time, the flow of the orbits is very sensitive to the positions of the current states and the direction of displacement thanks to forces of attraction and repulsion of their equilibrium points [10,16]. That is why important work has been presented with recurrent neural networks in identifying, characterizing, and predicting future states in chaotic dynamics. Consider the case of the work presented in [17], where predictions of regime changes and the durations of the orbits were made, in the chaotic trajectories of a system using an Eko state network (ESN). It achieved predictions of the future states of the system with accuracy in a few hundred steps of the iteration method. In this same sense, in [18], its authors proposed a feed-forward (multi-layer perceptron) neural network to classify the states of the Lorenz system between stable and unstable, and to improve with their results the automated decision-making while employing sequential decision making (SDM) framework.

The activation functions were used for the controlled application of different dynamic systems, such as the Gaussian or the Mexican hat; however, other works used the Morlet wavelet activation function for the estimation and prediction of time series [19]. Therefore, the Morlet wavelet activation function in this work is used to identify the dynamics of piecewise chaotic systems.

Recurrent wavelet first-order neural networks, or RWFONN, have been used for the implementation of neural controllers of electrical machines, emulated energy storage systems online, and identify the trajectories of the system. As in a recent work [20], the authors designed a super-twisting neural controller to emulate an energy storage system using an RWFONN for the identification of the states of the mathematical models of a permanent magnet synchronous machine (PMSM) and a direct current machine. They trained the neural network with the error filtering algorithm to adjust the synaptic weights of the network and guarantee that the tracking errors would converge to zero. Once the neural identification was achieved, they performed the synthesis of the controller through the proposed neural network structure, using the block control linearization technique and the super-twisting algorithm, thereby controlling the energy storage system emulated by the two electric machines.

In this work, we present a fast and adjustable alternative for the identification and reproduction of complex chaotic trajectories through an RWFONN. One advantage of using these first-order neural networks that adjust their synaptic weights thanks to their online training is the speed with which they work and identify the trajectory of the system, since they are trained with the current states and the error between the states of the neurons and the desired states of the system. They are an efficient alternative for the identification of complex systems that require real-time analysis and rapid adaptation to changes in their

trajectories [21]. In addition, it is intended to design a general first-order artificial neural network structure to identify different types of chaotic systems.

The rest of the article is divided as follows: in Section 2 we present the theory involved and particularities of the dynamic chaotic systems; Section 3 presents the specifications of the RWFONN; Section 4 describes the adjustments and settings of the network for identifying each of the chaotic systems; in Section 5 discussions on the results are presented. Finally, the main conclusions are presented in Section 6.

## 2. Chaotic Dynamical Systems

### 2.1. Simple Memristive Jerk System

The MSCS represents the behavior of a simple memristive jerk system described as one of the most algebraically simple chaotic trajectories [15]. It presents the following structure:

$$
\begin{aligned}
\dot{x_1} &= x_2, \\
\dot{x_2} &= x_3, \\
\dot{x_3} &= -x_3 - a x_3^2 - W(x_1)x_2 + b,
\end{aligned}
\tag{1}
$$

where $x_1$ corresponds to the internal memristor variable that stands for the magnetic flux, and $x_2$ and $x_3$ represent external variables. The flux control memductance is represented by $W(x_1) = 1.3x_1^2 - 1$, with $a, b \in \mathbf{R}$ commonly adjusted in a chaotic state by $a = 0.239$ and $b = 1$. The maximum Lyapunov exponent (MLE) of the system for these parameter values is 0.0529, calculated by the Wolf et al. algorithm presented in [22], proving that the system presents chaotic behavior.

The system results in a projection depicting a hidden attractor with no equilibria, see Section 4.2.

### 2.2. Unstable Dissipative System of Type I (UDSI)

The second system based on the jerky equations follows the same structure presented in [8,14], considering the class of piecewise linear systems given by:

$$
\dot{\mathbf{X}} = \mathbf{A}\mathbf{X} + \mathbf{B},
\tag{2}
$$

where $\mathbf{X} = [x_1, x_2, x_3]^T \in \mathbf{R}^3$ is the state vector, $\mathbf{B} \in \mathbf{R}^3$ represents an affine discrete real vector, and $\mathbf{A} = [a_{ij}] \in \mathbf{R}^{3 \times 3}$ with $i, j = 1, 2, 3$ denotes a nonsingular linear matrix. The equilibria of the systems are located at $\mathbf{X}^* = -\mathbf{A}^{-1}\mathbf{B}$. Additionally, as it is described in Definition 2.1 in [14], a system with a stability index of type I will be defined as a UDS type I. Furthermore, the following considerations must be fulfilled to determine that the system given in (2) is a UDS and it generates a scrolling attractor $\mathfrak{A}$.

1.  The linear part of the systems must be dissipative, satisfying $\sum_{i=1}^{3} \lambda_i < 0$, where $\lambda_i, i = 1, 2, 3$, are the eigenvalues of $\mathbf{A}$. Consider that an eigenvalue $\lambda_i$ must be negative real, and two $\lambda_i$ must be complex and conjugated with the positive real part $Re\{\lambda_i\} > 0$, resulting in an unstable equilibrium focus-saddle point $\mathbf{X}^*$. This equilibrium presents an stable manifold $M^s = span\{\lambda_1\} \in \mathbf{R}^3$ with a fast eigendirection and an unstable manifold $M^u = span\{\lambda_2, \lambda_3\} \in \mathbf{R}^3$ with a slow spiral eigendirection.
2.  The affine vector $\mathbf{B}$ must be considered as a discrete function that changes domains $\mathcal{D}_i \subset \mathbf{R}^3$ depending on where the trajectory is located. Thus, $\mathbf{R}^3 = \cup_{i=1}^{k} \mathcal{D}_i$.

Therefore, a hybrid system based on the continuous linear system given by (2) and the discrete function $\mathbf{B}$ will be given by:

$$\dot{\mathbf{X}} = \mathbf{A}\mathbf{X} + \mathbf{B}(\mathbf{X}),$$

$$\mathbf{B}(\mathbf{X}) = \begin{cases} \mathbf{B}_1, & \text{if } \mathbf{X} \in \mathcal{D}_1; \\ \mathbf{B}_2, & \text{if } \mathbf{X} \in \mathcal{D}_2; \\ \vdots & \vdots \\ \mathbf{B}_k, & \text{if } \mathbf{X} \in \mathcal{D}_k. \end{cases} \tag{3}$$

Now, to generate hybrid multi-scroll systems, the matrix $\mathbf{A}$ and the affine vector $\mathbf{B}$ can be given by:

$$\mathbf{A} = \begin{pmatrix} 0 & 1 & 0 \\ 0 & 0 & 1 \\ -1.5 & -1 & -1 \end{pmatrix}, \mathbf{B} = \begin{pmatrix} 0 \\ 0 \\ b \end{pmatrix}; \tag{4}$$

resulting in the following eigenvalues: $\Lambda = \{-1.20, 0.10 \pm 1.11i\}$. The system (3) has its equilibria at $\mathbf{X}^* = -\mathbf{A}^{-1}\mathbf{B}(\mathbf{X}) = [2/3b, 0, 0]^T$. Therefore, the component $b$ of $\mathbf{B}$ is governed by a commutation control law to generate 2 scrolls, depending on the value that the state $x_1$ takes in the following manner:

$$b(x_1) = \begin{cases} 2, & \text{if } x_1 \geq 1; \\ 1, & \text{otherwise.} \end{cases} \tag{5}$$

In Figure 1, the projection of the system (2) with (4) and (5) onto the $(x_1, x_2)$ plane is depicted, resulting in a two-scroll attractor oscillating around the equilibria of the system depicted in red asterisks. The red lines correspond to the stable fast eigendirections, and the blue ones stand for the unstable slow eigendirections. Notice that the displacement of the scrolls occurs along the $x$ axis, as the commutation surface and equilibria determine. The black vertical slashed line represents the position of the commutation surfaces given in (5), depicting the two existing domains (i.e., left and right) of the system and resulting in a scroll for the attractor on each one of them $\mathbf{R}^3 = \cup_{i=1}^{2} \mathcal{D}_i$.

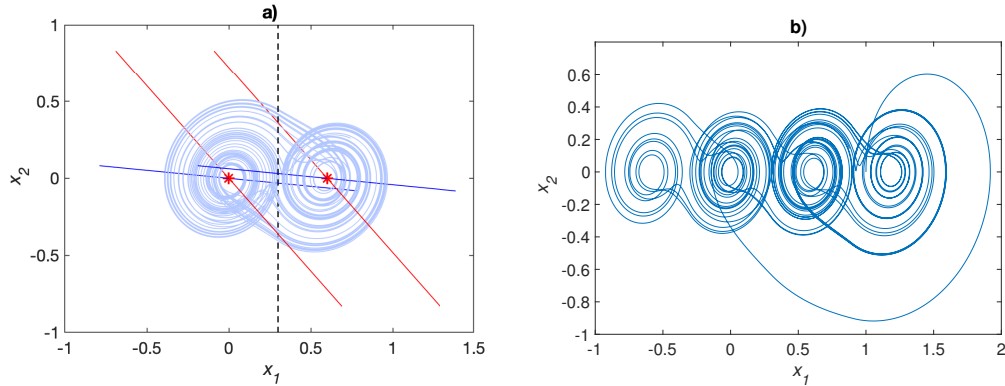

**Figure 1.** Projection of the system (2) with (4) and the commutation laws given in: (**a**) (5); (**b**) (6)—onto the $(x_1, x_2)$ plane. Both were initialized at $\mathbf{X}^0 = [1, 0, 1]^T$.

Now, the numbers of domains and resulting scrolls in the attractor can be adjusted by a specific commutation law. Consider the following commutation control law:

$$b(x_1) = \begin{cases} 1.8, & \text{if } x_1 \geq 0.9; \\ 0.9, & \text{if } 0.3 \leq x_1 < 0.9; \\ 0, & \text{if } -0.3 \leq x_1 < 0.3; \\ -0.9, & \text{otherwise.} \end{cases} \tag{6}$$

In this case, the system presents 4 domains $\mathbf{R}^3 = \cup_{i=1}^{4} \mathcal{D}_i$ for its four equilibrium points. Therefore, the attractor of the system will result in the following 4-scroll projection,

as depicted in Figure 1, which presents a projection of the system (2) with (4) and (6) onto the $(x_1, x_2)$ plane. The maximum Lyapunov exponent (MLE) of the system for these parameter values and commutation law is $MLE = 0.1020$, proving that the system presents chaotic behavior.

## 3. Materials and Methods

### 3.1. Recurrent Wavelet First-Order Neural Network

In the recent works [20,23], the authors controlled different systems through a RW-FONN, where they showed the efficiency of the artificial neural network. The general structure of the system is given by

$$\dot{y}_j^i = -\alpha_j^i y_j^i + (w_{jk}^i)^\top \psi_{jk}^i, \tag{7}$$

where $y_j^i$ is the state of the $i$-th neuron; $\alpha_j^i > 0$ for $i = 1, 2, \ldots, n$ is part of the underlying network architecture, and it is fixed during the training process; $w_{jk}^i$ is the $k$-th adjustable synaptic weight connecting the $j$-th state to the $i$-th neuron; and $\psi_{jk}^i$ is a Morlet wavelet activation function defined by $\psi(\chi) = e^{(-\chi^2/\beta)}\cos(\mu\chi)$, where $\chi$ is the state of the original system to identify; the parameters $\beta$ and $\mu$ are the expansion and dilation terms.

The systems (1) and (2) were identified online using the RWFONN, where the synaptic weights were adjusted via the filtered error algorithm.

### 3.2. Filtered Error Algorithm

The identification scheme starts from the differential equation that describes the unknown system:

$$\dot{\chi}_j^i = -\alpha_j^i \chi_j^i + (w_{jk}^i)^{*\top} \psi_{jk}^i. \tag{8}$$

Based on (8), the identifier can be chosen as

$$\dot{y}_j^i = -\alpha_j^i y_j^i + (w_{jk}^i)^\top \psi_{jk}^i. \tag{9}$$

In this way, the identification error is defined as $\xi_j'^i = y_j^i - \chi_j^i$ such that

$$\begin{aligned}
\dot{\xi}_j'^i &= \dot{y}_j^i - \dot{\chi}_j^i \\
&= -\alpha_j^i y_j^i + (w_{jk}^i)^\top \psi_{jk}^i - (-\alpha_j^i \chi_j^i + (w_{jk}^i)^{*\top} \psi_{jk}^i) \\
&= -\alpha_j^i y_j^i + (w_{jk}^i)^\top \psi_{jk}^i + \alpha_j^i \chi_j^i - (w_{jk}^i)^{*\top} \psi_{jk}^i \\
&= -\alpha_j^i(y_j^i - \chi_j^i) + (w_{jk}^{i\top} - w_{jk}^{i*\top})\psi_{jk}^i.
\end{aligned} \tag{10}$$

Equation (10) can be rewritten as

$$\dot{\xi}_j'^i = -\alpha_j^i \xi_j^i + \widetilde{w}_j^i \psi_{jk}^i, \tag{11}$$

where $\widetilde{w}_j^i = w_j^i - w_j^{*i}$. The synaptic weights $w_j^i$ for $i = 1, 2, \ldots, n$ are adjusted according to the learning law [24]:

$$\dot{w}_j^i = -\gamma_j^i \psi_{jk}^i \xi_j'^i \tag{12}$$

called "filtered error".

**Theorem 1.** *Consider the RWFONN model whose weights are adjustable according to (12) for each $i = 1, 2, \ldots, n$, so that*

1.　$\xi_j'^i$, $w_j^i \in \mathcal{L}_\infty$ *(i.e., $\xi_j'^i$ and $w_j^i$ are uniformly bounded);*
2.　$\lim_{t \to \infty} \xi_i(t) = 0$.

**Proof.** Proof see [24,25]. □

In Appendix A, we present the boundedness of the identification error $\xi_j'^i$ given by the synaptic weights $w_j^i$.

## 4. Neural Identification Results

In order to identify the UDSI and MSCS dynamical states, we propose a RWFONN with the following structure:

$$
\begin{aligned}
\dot{y}_1 &= -\alpha_1 y_1 + \delta_1 w_1 \psi_1(\chi_1) + y_2, \\
\dot{y}_2 &= -\alpha_2 y_2 + \delta_2 w_2 \psi_2(\chi_2) + y_3, \\
\dot{y}_3 &= -\alpha_3 y_3 + \delta_3 w_3 \psi_3(\chi_3) + u.
\end{aligned}
\tag{13}
$$

It is worth mentioning that $u$ is the same input ($b$) of the chaotic systems to be identified. In this section, we present the results of the numerical simulation of the neural identification of the systems MSCS and UDSI (1) and (2), through the neural network structure (13). The simulations were performed using Matlab/Simulink (MatlabTM) with a Runge–Kutta algorithm with a 0.01 step size.

### 4.1. Analysis of Identification via the Euclidean Distance between Trajectories

The identification of the system through the neural network can be further analyzed by computing the Euclidean distance between trajectories. This distance has been defined as follows:

$$
d(t) := \sqrt{(d_1(t_i))^2 + (d_2(t_i))^2 + (d_3(t_i))^2},
\tag{14}
$$

where $d_1(t_i) = (x_1(t_i) - y_1(t_i)), d_2(t_i) = (x_2(t_i) - y_2(t_i)), d_3(t_i) = (x_3(t_i) - y_3(t_i))$, and $t_i$ corresponds to each time step iterated using numerical integration. In this case, a value of $d(t) = 0$ corresponds to a synchronous solution, whereas $d(t) \neq 0$ corresponds to uncorrelated behavior.

### 4.2. Neural Identification for MSCS

For neural identification of the MSCS, the parameter values were adjusted to $\alpha_1 = \alpha_2 = \alpha_3 = \delta_1 = \delta_2 = \delta_3 = 5$; the filtered error parameters to $\gamma_1 = \gamma_2 = \gamma_3 = 5000$; the parameters of Morlet wavelet activation functions to $\mu_1 = \mu_2 = \mu_3 = 0.01$ and $\beta_1 = \beta_2 = \beta_3 = 100$. Figure 2a–c show the neural identification of states of the MSC. In Figure 2a, the identification of state variable $x_1$ is the red dashed line, and the solid blue line represents the state of the RWFONN ($y_1$). To show the convergence of these results, the initial conditions are given as $x_1 = 0$ and $y_1 = 0.2$. Note that in the detail of the figure, the identification convergence is given as 0.2 s, approximately. Table 1 shows the neuronal identification results for the variables $x_2$ (see Figure 2b) and $x_3$ (see Figure 2c), where we present the initial conditions for each variable and the convergence time. It is worth mentioning that the simulations were carried out with time in seconds but divided into step sizes of 0.01 s.

**Table 1.** Neural identification results for $x_2$ and $x_3$ states of MSCS.

| Figure | State | Line | Initial Condition | Convergence |
|---|---|---|---|---|
| Figure 2b | $x_2$ and $y_2$ | Red dashed and blue continuous | $x_2 = -2$ and $y_2 = -2.1$ | 0.2 s |
| Figure 2c | $x_3$ and $y_3$ | Red dashed and blue continuous | $x_3 = -2$ and $y_3 = -1.9$ | 0.3 s |

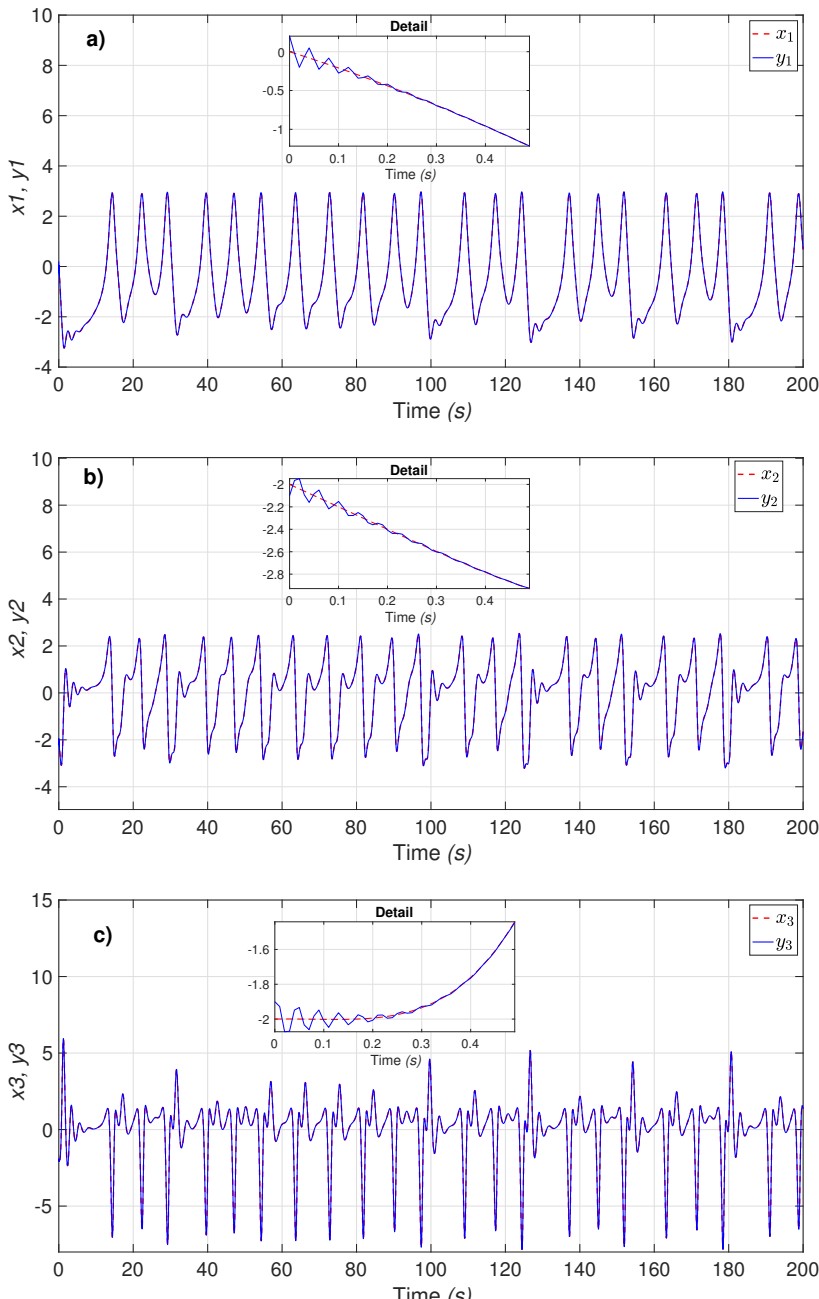

**Figure 2.** States of the MSCS given by (1) and the neural identification in (13) over the iterated time for the following states: (**a**) $x_1$ and $y_1$ behavior. (**b**) $x_2$ and $y_2$. (**c**) $x_3$ and $y_3$. All the graphics have zooms to appreciate the convergence times of the states of the neural identification and the states of the system.

Figure 3a shows the projection of the state spaces in the $(x_1, x_2)$ planes from the MSCS given by Equation (1), and Figure 3b shows the projection of the $(y_1, y_2)$ plane with the state variables of the RWFONN in Equation (13). In addition, the Euclidean distance between the states of the system and the RWFONN according to Equation (14) is presented in Figure 3c for a range of time between $0 \leq t_i \leq 200$. Notice that after a brief period, the value of $d(t)$ drops to almost zero. However, it presents small perturbations of length $d(t_i > 10) < 0.02$, proving that the identification made by the RFWONN is almost identical.

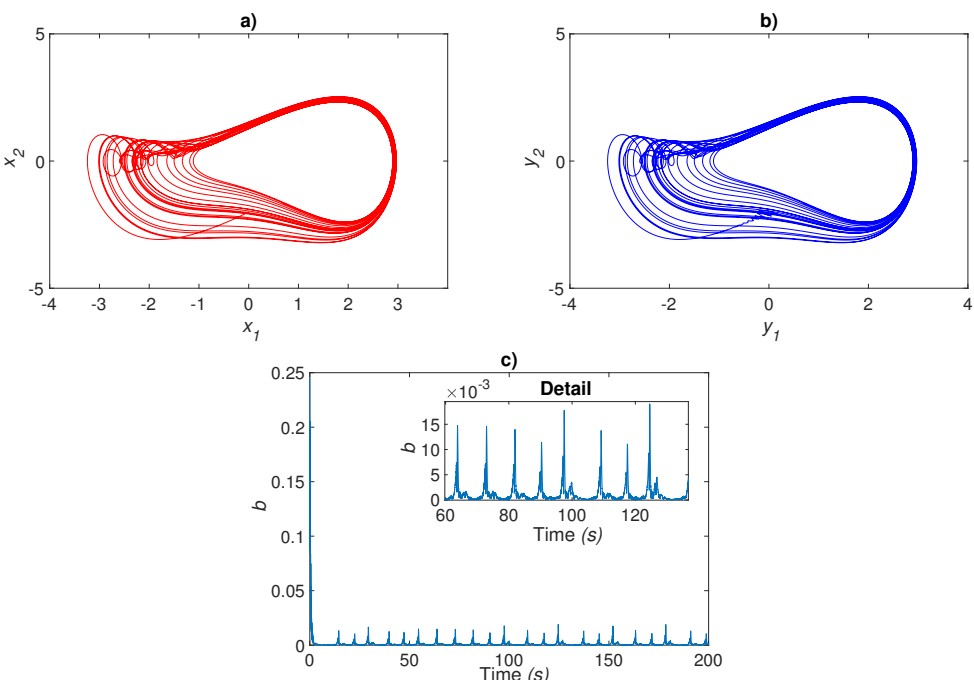

**Figure 3.** Projection of the system MSCS and RWFONN: (**a**) MSCS state space: $x_1$ and $x_2$; (**b**) RWFONN state space: $y_1$ and $y_2$. (**c**) Euclidian distance between the states of the system and the RWFONN, as presented in Equation (14).

### 4.3. Neural Identification for UDSI

The neuronal identification for UDSI was obtained using the same neural network structure (13) and parameters. In Figure 4a, the identification of state variable $x_1$ of USDI is the red dashed line, and the blue solid line represents the state of the RWFONN ($y_1$). To show the convergence of these results, the initial conditions are given as: $x_1 = 1$ and $y_1 = 0.9$. Note that in the detail of the figure, the identification convergence is given in 0.4 s, approximately. This can be better appreciated with the Euclidean distance between the states of the system and the RWFONN according to Equation (14), as is depicted in Figure 5c for a range of time between $0 \leq t_i \leq 50$. Notice that after a brief period, the value of $d(t)$ drops to almost zero, nearly 2 s. After this time, the size of the distance remains $d(t_i >) < 5 \times 10^{-4}$, proving that the identification made by the RFWONN is almost identical.

Table 2 shows the neuronal identification results for the variables $x_2$ (see Figure 4b) and $x_3$ (see Figure 4c) of UDSI, where we present the initial conditions for each variable and the convergence time.

**Table 2.** Neural identification results for $x_2$ and $x_3$ states of UDSI.

| Figure | State | Line | Initial Condition | Convergence |
|---|---|---|---|---|
| Figure 4b | $x_2$ and $y_2$ | Red dashed and blue continuous | $x_2 = 0$ and $y_2 = 0.1$ | 0.4 s |
| Figure 4c | $x_3$ and $y_3$ | Red dashed and blue continuous | $x_3 = 1$ and $y_3 = 1.1$ | 0.3 s |

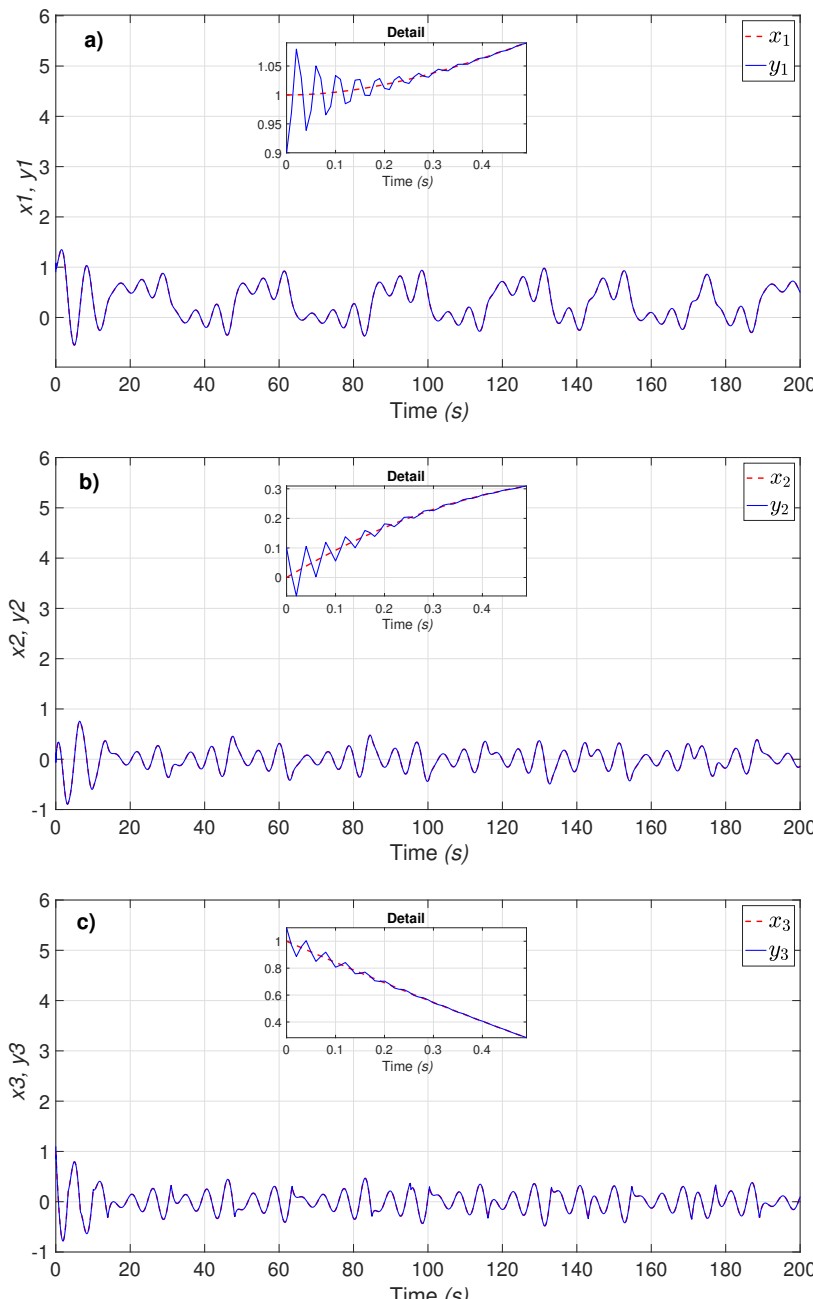

**Figure 4.** States of the UDSI given by (2) with (4) and (6) and the neural identification in (13) over the iterated time for the following states: (**a**) $x_1$ and $y_1$ behavior. (**b**) $x_2$ and $y_2$. (**c**) $x_3$ and $y_3$. All the graphics present zooms with more detail to appreciate the convergence time of the states of the neural identification and the states of the system.

Figure 5 shows the projection of the state spaces in the $(x_1, x_2)$ planes from the UDSI, and in the $(y_1, y_2)$ plane with the state variables of the RWFONN, respectively.

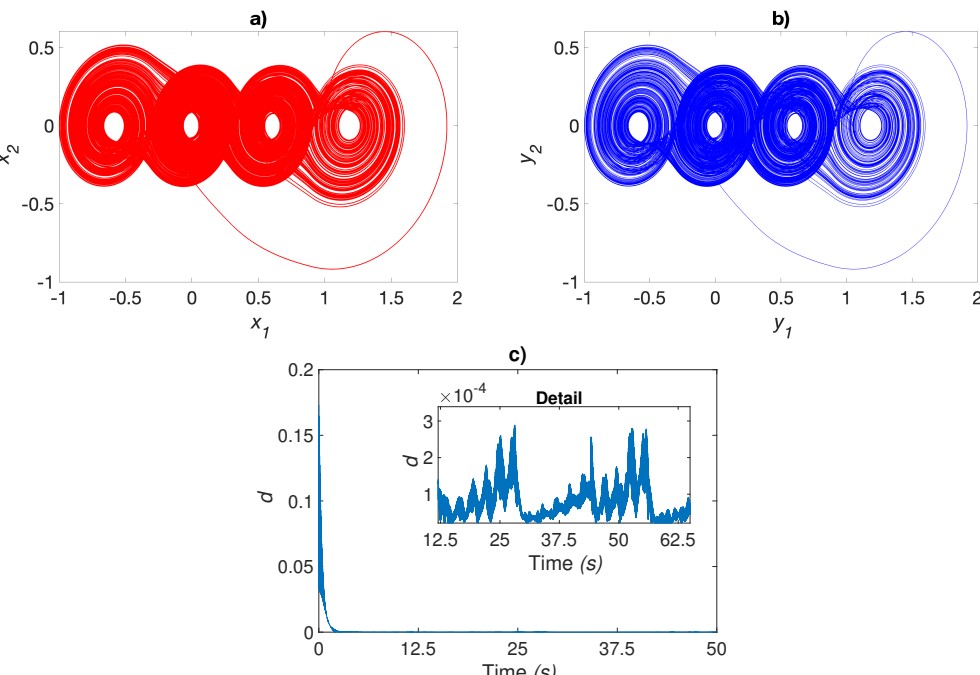

**Figure 5.** Projection of the system USDI and RWFONN: (**a**) UDSI state space: $x_1$ and $x_2$; (**b**) RWFONN state space: $y_1$ and $y_2$. (**c**) Euclidian distance between the states of the system and the RWFONN.

## 5. Discussion

Regarding the parameters used for the RWFONN in the identification of the MSCS and UDSI chaotic systems: The values presented in Sections 4.2 and 4.3 were chosen considering the study of the maximum Euclidean distance discussed in Appendix B. Notice that there is a slight difference in the simulation results of the Euclidean distance presented in Figures 3 and 5; notwithstanding, the error is very low. This may be caused by the difference in the equations of the systems and the parameter selection carried out in the activation function, which can be further adjusted for an improvement in the results. In this light, in a recent work [23], an artificial neural network structure, RWFONN, was proposed to identify and control an Erbium-doped fiber laser system which also presents multistable chaotic behavior (similarly to the UDSI system). Therefore, we can mention that the network's parameters can be adjusted for greater accuracy in the neuronal identification process, as long as the error is low and the resulting values of the filtered error algorithm in (12) remain considerably low depending on the required application.

## 6. Conclusions

The results of the simulation in this work show the neural identification of the state variables of the MSCS and UDSI chaotic systems through the proposed artificial neural network. It should be noted that there are works that apply neural networks to approximate mathematical models, but here we tested the application of a new neural network, specifically, a RWFONN, on a model of unstable chaotic systems. The numerical results showed good identification of the states of systems (1) and (2), due to the convergence time. Furthermore, as future work, it is intended to implement the RWFONN for electronic circuits, to predict regime changes and generate hyperchaotic multi-scroll systems. Based on the results obtained from neural identification with the same network structure, we can mention that a general structure has been designed to identify different chaotic systems, using only three neurons in the artificial neural network, since it was not necessary to include more neurons in the neural identification of systems (1) and (2), because if more neurons are used in artificial neural networks to achieve neuronal identification, a higher computational cost is required to achieve it.

Finally, due to the numerical results presented here and the results mentioned in the works that were referenced in Section 3, this approximation is useful in the study of other types of physical systems, in which the model equations are not necessary—only the measurable or observable variables that they might present are necessary.

**Author Contributions:** Conceptualization, D.A.M.-G., L.J.O.-G., J.H.G.-L., G.H.-C. and C.S.-M.; investigation, D.A.M.-G. and L.J.O.-G.; methodology, D.A.M.-G. and L.J.O.-G.; supervision, L.J.O.-G.; validation, L.J.O.-G.; writing—original draft, L.J.O.-G.; writing—review and editing, D.A.M.-G., J.H.G.-L., G.H.-C. and C. S.-M. All authors have read and agreed to the published version of the manuscript.

**Funding:** This research received no external funding.

**Data Availability Statement:** The data of the article was simulated numerically by Matlab codes. These files can be downloaded from http://a.uaslp.mx/axiom2168578 (accessed on 5 February 2023). Also, the graphical video abstract of the article can be found in the following link: https://youtu.be/vWuonhdNHNo (accessed on 5 February 2023).

**Acknowledgments:** The first author acknowledges the support of the CONACyT, being benefited with an academic postdoctoral fellowship with application number: 2290436. G.H.C. acknowledges the State Council of Science and Technology of Jalisco (COECYTJAL), project number 10304-2022, with title "Control difuso aplicado a sistemas de tecnología solar térmica y fotovoltaica para implementación de planta piloto para pasteurización de leche" de la Convocatoriadel Fondo de Desarrollo Científico de Jalisco para Atender Retos Sociales "FODECIJAL 2022".

**Conflicts of Interest:** The authors declare no conflict of interest.

## Appendix A. Identification Error Boundedness

Suppose that systems (1) and (2), and further, model (7), are initially in the same state, $y(0) = \chi(0)$. Then, for any $\epsilon > 0$ and any finite $T > 0$, there exists an integer $L$ and a matrix $w* \in \mathbb{R}^{L \times n}$ such that the state $y(t)$ of the RWFONN model (7) and weight values $w = w*$ satisfy

$$\sup_{0 \leq t \leq T} |y(t) - \chi(t)| \leq \epsilon.$$

Next, using the Bellman–Gronwall Lemma [26], the identification error $\xi'^{i}_{j} = y^{i}_{j} - \chi^{i}_{j}$ is bounded by

$$\|\xi'\| \leq \frac{\epsilon}{2}. \tag{A1}$$

**Proof.** See reference [27]. $\square$

## Appendix B. Network Parameter Adjustment

To determine the values of the parameters of the RWFONN $\alpha_{1,2,3}, \delta_{1,2,3}$, the filtered error parameters $\gamma_{1,2,3}$, and the parameters of the Morlet wavelet activation functions $\mu_{1,2,3}$ and $\beta_{1,2,3}$, the following study was performed. Variation of the parameter was implemented to determine the maximum Euclidean distance, as presented in (14) for each corresponding value. In Figure A1 are the results for the MSCS system and the RWFONN. First, in Figure A1a, the values of the parameter are $0.25 \leq \alpha_{1,2,3} \leq 50$. In Figure A1b, they are $100 \leq \gamma_{1,2,3} \leq 1 \times 10^4$. In Figure A1c, they are $0 < \mu_{1,2,3} \leq 1$. In Figure A1d, they are $10 \leq \beta_{1,2,3} \leq 500$. Notice that for the selected values presented in Section 4.2, the graphs depict a low Euclidean distance.

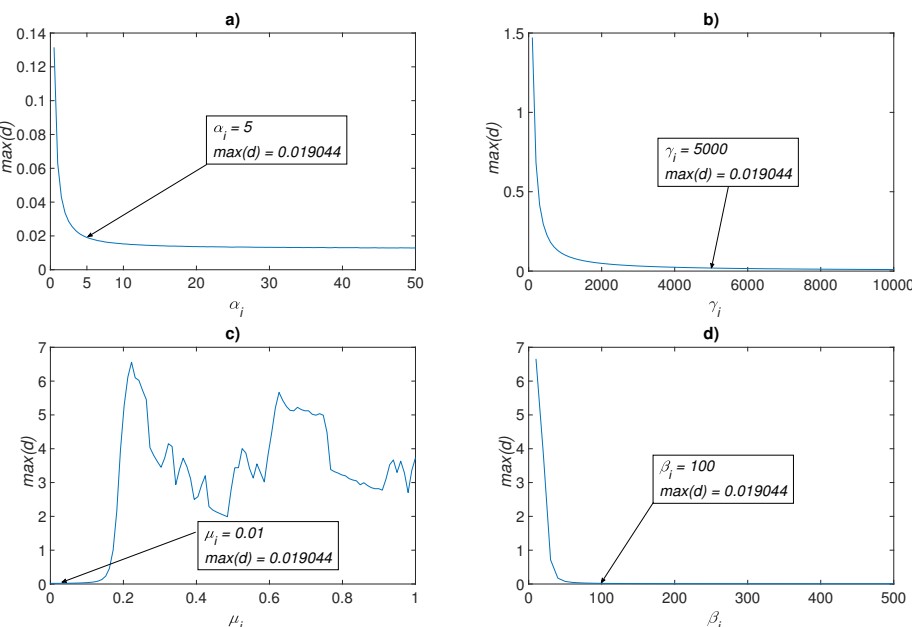

**Figure A1.** Maximum Euclidean distance between the states of the MSCS and the RWFONN, as presented in Equation (14) for the variations in the following network parameters: (**a**) $0.25 \leq \alpha_{1,2,3} \leq 50$. (**b**) $100 \leq \gamma_{1,2,3} \leq 1 \times 10^4$. (**c**) $0 < \mu_{1,2,3} \leq 1$. (**d**) $10 \leq \beta_{1,2,3} \leq 500$.

A similar case is presented in Figure A2, as it is depicts the results for the UDSI system and the RWFONN for the same parameters.

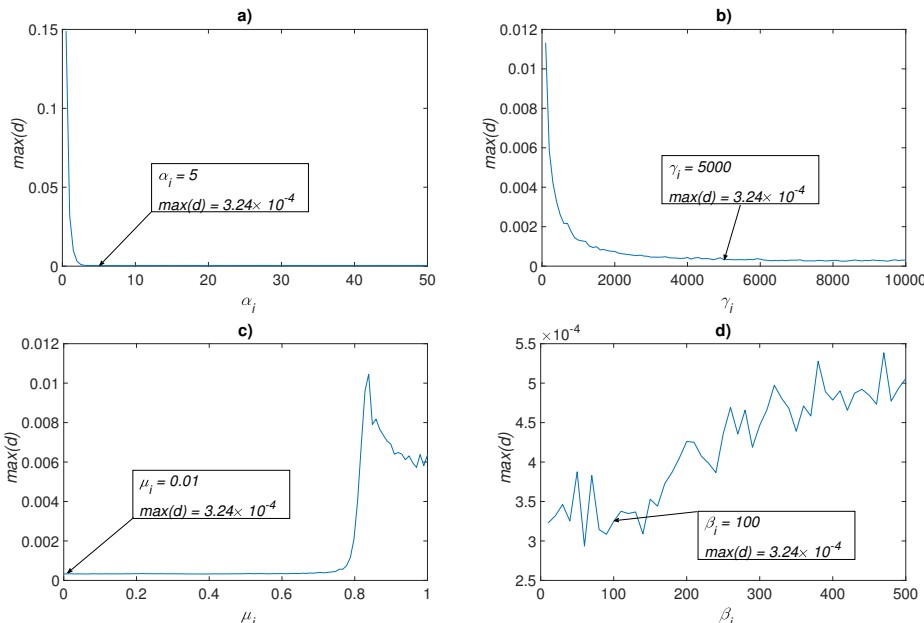

**Figure A2.** Maximum Euclidean distance between the states of the UDSI and the RWFONN, as presented in Equation (14), for the variations of the following network parameters: (**a**) $0.25 \leq \alpha_{1,2,3} \leq 50$. (**b**) $100 \leq \gamma_{1,2,3} \leq 1 \times 10^4$. (**c**) $0 < \mu_{1,2,3} \leq 1$. (**d**) $10 \leq \beta_{1,2,3} \leq 500$.

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
