# Peer review of "Identification of Chaotic Dynamics in Jerky-Based Systems by Recurrent Wavelet First-Order Neural Networks with a Morlet Wavelet Activation Function"

_axioms, doi:10.3390/axioms12020200_

Round 1

Reviewer 1 Report

This paper presents a strategy for identifying and reproducing complex chaotic trajectories based on Recurrent Wavelet First Order Neural Network (RWFONN). By trained online and adjusting parameters, the RWFONN can identify state variables of MSCS and UDSI chaotic systems more accurately. The results are proved via numerical simulation. But the following comments need to be addressed.

1. The summary description of the study results should be given in the abstract to intuitively demonstrate the advantages of the study.

2. Can the strategy for identifying and reproducing chaotic dynamics be applied to neuron models. Please discuss in Introduction and Conclusion.

3. There should not be a line feed between line 109 and line 110.

4. Equation (13) should end with “.” instead of “,”.

5. The sentences in line 195 “Figure 3 a) show the projection…” should be modified as “Figure 3 a) shows the projection…”. The sentences in line 196 “and Figure 3 b) the projection of the (y1, y2)…” should be modified as “and Figure 3 b) shows the projection of the (y1, y2)…”.

6. In Figures 2 and 4, the labels of coordinates are suggested to be supplemented.

7. Some labels of abscissa are not uniform, such as “Time (s)” and “Time (s) (italic)”, which should be modified properly.

8. The paper has some language issue which needs to be checked and corrected in the revision.

Author Response

Thank you for evaluating our work and for your valuable recommendations. Please see our answers to your comments in the attached PDF file.

Reviewer 2 Report

This work deals with two dynamic chaotic systems whose equations are based on the jerk systems. The authors stated that system  (1) has a fixed point 

(0,0,0). However,  this is wrong  since the third  equation of (1)has the constant term b and b does not equal to zero. 

We need to check the  numerical results of systems (2,3,...). Could you please ask the authors to send you the codes which are used in the numerical calculations. Please send me those codes. 

Author Response

(The authors gave the same response as above.)

Reviewer 3 Report

Title: Identification of chaotic dynamics in jerky based systems by recurrent wavelet first order neural networks with a Morlet wavelet activation function
Authors: D. A. Magallón-García , L. J. Ontanon-Garcia * , J. H. García-López , G. Huerta-Cuéllar , C. Soubervielle-Montalvo

This paper is devoted to the description of a new neural network using a recurrent first-order wavelet neural network based on unstable chaotic systems. Using an error filtering algorithm, a neural network was trained when the activation function is a Morlet wavelet. The synaptic weights are adjusted via the filtered error algorithm.  The correctness of this approach is studied and demonstrated on the basis of numerical modeling.

I believe that the manuscript brings new value to the topic of neural networks and can play the important role in the future development. That's why it deserves to be published in the Axioms. However, I have a couple of small questions to the authors that would allow them to further improve the manuscript.

1)The structure of the neural network you have developed is based on only three neurons. Explain whether it is supposed to scale the number of neurons and how their number can affect the operation of the network, the processes of setting parameters and weights?

2) In the conclusions, it would be necessary to describe more specifically the range of tasks for which the proposed neural network would be useful, to make a comparison with existing analogues.

Author Response

(The authors gave the same response as above.)

Round 2

Reviewer 1 Report

This paper can be accepted in the current version.

Reviewer 2 Report

Now is it OK for Publication